# DETECTING CHANGE POINTS IN TIME SERIES VIA CURVATURES OF REPRESENTATION TRAJECTORIES

## ABSTRACT

*Change points* are the timestamps at which a time series experiences meaningful changes. Recently, representation-based change point detection has gained popularity, but its emphasis on consecutive distance difference backfires, especially when the changes are gradual. In this paper, we propose a change point detection method, **RECURVE**, based on a novel change metric, the ***curvature*** of a representation trajectory, to accommodate both gradual and abrupt changes. Here, a sequence of representations in the representation space is interpreted as a trajectory, and a curvature at each timestamp can be computed. Using the theory of random walk, we formally show that the mean curvature is lower near change points than at other points. Extensive experiments using diverse real-world time-series datasets confirm the superiority of RECURVE over state-of-the-art methods.

## 1 INTRODUCTION

In a time series composed of sequential data points (simply points) indexed by timestamps, there are *change points* signifying transitions between different classes or states, such as a shift from running to walking (Aminikhanghahi & Cook, 2017; Truong et al., 2020). Detecting change points is a crucial task in preprocessing and diverse applications of time-series data. As preprocessing, they partition a time series into segments of coherent points, accelerating annotation of the time-series for further analysis and giving additional supervision in classification (Li et al., 2021; Ishikawa et al., 2021). As primary tasks, they are valuable for identifying changes that require human attention in a variety of domains, including climate, health care, finance, and manufacturing; epilepsy detection, stock price tracking, and action segmentation are examples of possible applications (Reeves et al., 2007; Malladi et al., 2013; Pepelyshev & Polunchenko, 2016; Xia et al., 2020).

Representation-based change-point detection methods (Ryck et al., 2021; Deldari et al., 2021) are prevalent today because they do not require specific assumptions on time-series properties, such as distribution or temporal shape, and can handle high dimensionality due to the capability of a self-supervised model that autonomously learns distinctive features from raw time series. In these methods, a self-supervised model (Tonekaboni et al., 2021; Yue et al., 2022; Zhang et al., 2022) is first used to derive a representation of each point, and then a point is identified as a change point if its representation significantly deviates from those of adjacent points. Let's refer to the points close to a change point as *inter-segment* points and the remaining points as *intra-segment* points. In short, these methods operate by assuming that the *distance* between consecutive representations is greater between inter-segment points than between intra-segment points.

However, this assumption on the distance difference does *not* always hold, especially when the changes are subtle or gradual. Time-series representation learning methods often pursue preserving the *temporal coherence* of a time series as their training goal is to make temporally close points similar in their representations and distant points dissimilar (Tonekaboni et al., 2021) As demonstrated in Figure 1, the consecutive distances are not clearly distinguishable between intra- and inter-segment points for relatively subtle changes with `stair up` ↔ `stair down` because just the direction of motion differs between the two classes, whereas they are for abrupt changes with `stand` ↔ `sit`. Thus, the inability to handle subtle changes hinders achieving an overall good performance.

In this work, we take a novel perspective on detecting change points by leveraging ***curvatures*** instead of distances in the representation space. As shown in Figure 2, the curvature at a point in a curve measures the instantaneous rate of direction change, or more precisely, the amount by which

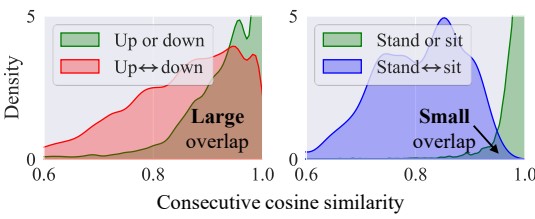
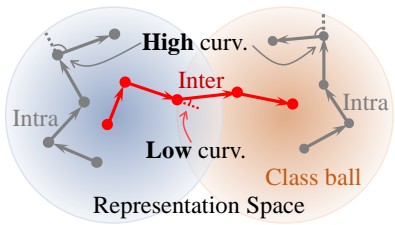

Figure 1: Consecutive distance (cosine similarity) distribution from intra- and inter-segment points in the representations of the HAPT dataset.

Figure 2: Curvature comparison between intra- and inter-segment points in a representation space.

the curve deviates from being a straight line (Lewiner et al., 2005). Suppose that a sequence of point representations from a time series constitutes a *representation trajectory*. We observe that, regardless of whether the changes are gradual or abrupt, the direction of the representation trajectory tends to change *more sharply* (showing a *higher curvature*) at intra-segment points than at inter-segment points. Accordingly, we contend that the curvature of a representation trajectory should be a very promising indicator for change point detection.

Using Figure 2, we justify the intuition behind *curvature-based* change point detection. Because representation learning tries to learn class-separated features, well-embedded points of a certain class (or a segment) can be drawn from its class-specific ball (Wang et al., 2022; Parulekar et al., 2023). That is, the representation trajectory of intra-segment points is confined within a class ball, whereas that of inter-segment points is not. Then, for intra-segment points to reside exclusively within a class ball, their representation trajectory needs to make sharp turns frequently. In contrast, the transition from one class ball to another does not necessarily make sharp turns. This observation is formally proven by the relationship between the mean curvature and the radius of a confining hypersphere, assuming a random walk of a point representation (see Section 3.4).

Overall, for change point detection, treating a sequence of representations as a trajectory and measuring its curvature is an entirely novel approach, which results in RECURVE (Representation trajEctory CURVaturE). A representation trajectory is derived by a time-series representation learning method, and the curvature at each point is calculated very efficiently; then, the points whose curvature is relatively small are identified as change points. RECURVE is simple yet powerful, and can be combined with any time-series representation learning method. We conduct comprehensive evaluations on a variety of time-series datasets, comparing it against state-of-the-art change point detection methods. The results demonstrate that RECURVE consistently enhances the accuracy, achieving improvements of up to 12.7%. Furthermore, this superiority is shown to exist regardless of the degree of change between different classes.

## 2 RELATED WORK

### 2.1 CHANGE POINT DETECTION

Time-series change point detection methods assess the dissimilarity between two successive intervals and apply a threshold to pinpoint the positions of change points. There are multiple methods available for quantifying dissimilarity: **(1)** conducting statistical tests, **(2)** quantifying the deviation from discovered temporal patterns, and **(3)** calculating distances between the representations learned from a self-supervised model. We summarize each category here, with additional in-depth details available in extensive surveys (Aminikhanghahi & Cook, 2017; Truong et al., 2020).

Statistical tests often rely on the probability density ratio of two consecutive intervals as a key statistic. CUSUM is a traditional parametric algorithm that adds up the log likelihood ratio when a probability density function is given (Basseville & Nikiforov, 1993; Jeske et al., 2009; Cho & Fryzlewicz, 2015). RuLSIF is a non-parametric algorithm that directly estimates the probability density ratio using Pearson divergence without a probability density function (Yamada et al., 2013; Feuz et al., 2014; Hushchyn & Ustyuzhanin, 2021). A kernel-based statistical test maps each interval to a kernel space and then computes the kernel Fisher discriminant ratio as a statistic (Harchaoui et al., 2008; 2009). KL-CPD uses a deep neural network as a generator for kernel parameters, which solves high sensitivity in selecting parameters (Chang et al., 2019).

The proactive discovery of frequent temporal patterns is necessary for temporal pattern-based change point detection. FLOSS stores the locations of similar subsequences in a time series using

Matrix Profile and measures the likelihood of a regime change (Gharghabi et al., 2019). Motif-based change point detection relies on the identification of short temporal patterns (motifs) determined through the minimum description length criterion; these motifs are then compared for similarity with other subsequences within a time series (Zakaria et al., 2012; Xia et al., 2020). ESPRESSO, on the other hand, is a hybrid of pattern- and statistic-based approaches, detecting a wide range of change points across different scenarios and data types (Deldari et al., 2020).

Representation-based change point detection methods are distinguished by the manner in which a self-supervised model is trained. TIRE exploits an autoencoder to retain time-invariant features in consecutive timestamps to make representations of change points salient (Ryck et al., 2021); after training, the output representations undergo a process of smoothing, wherein a moving average is applied prior to the dissimilarity computation. TS-CP$^2$ leverages contrastive learning techniques to promote close proximity between representations of two consecutive timestamps and distant proximity between representations at randomly selected timestamps (Deldari et al., 2021); it examines the difference between each consecutive distance and the moving average.

## 2.2 TIME-SERIES REPRESENTATION LEARNING

Time-series representation learning builds a model to create versatile representations capable of performing diverse downstream tasks such as classification, forecasting, and anomaly detection (Zhang et al., 2023; Ma et al., 2023). Reconstruction-based learning methods train autoencoder-based deep neural networks using a reconstruction loss. TimeNet is an early example that uses a sequence-to-sequence autoencoder and uses the hidden embedding extracted from the encoder as a representation (Malhotra et al., 2017). DTCR extends traditional reconstruction-based learning by incorporating a $k$-means loss alongside the reconstruction loss (Ma et al., 2019). Input masking is also commonly used for reconstructing data with specific timestamps intentionally masked or hidden (Shao et al., 2022; Chowdhury et al., 2022; Chauhan et al., 2022).

In contrastive learning, the Info-NCE (Noise Contrastive Estimation) loss plays a pivotal role by bringing a positive pair closer together and pushing a negative pair apart in the representation space. An early approach considers a sampled window and a subsequence from the window as a positive pair (Franceschi et al., 2019). In recent methods such as TNC (Temporal Neighborhood Coding), the temporal distance serves as a criterion for identifying a positive pair, keeping two neighboring timestamp representations close (Tonekaboni et al., 2021; Deldari et al., 2021; Chen et al., 2022). Following the principles of SimCLR (Chen et al., 2020), a positive pair can be created by pairing a sampled window with its augmentation which involves data perturbation or context changes (Eldele et al., 2021; Yue et al., 2022). Besides, the Fourier transform of a time series serves as an augmentation technique for generating positive pairs or providing a new representation space (Woo et al., 2021; Yang & Hong, 2022; Zhang et al., 2022).

## 3 RECURVE: CURVATURE-BASED CHANGE POINT DETECTION

### 3.1 PRELIMINARIES AND PROBLEM SETTING

**Dataset and Model:** Let $\mathcal{X} = (\mathbf{x}_t)_{t=1}^T$ be a time series, where $T$ is the total number of points, and $\mathbf{x}_t \in \mathbb{R}^d$ is a $d$-dimensional point at timestamp $t$. Let $\mathcal{C} = \{t_k \mid k \in [\![1, K]\!]\}$ be a set of the timestamps for the ground-truth change points. Considering class labels annotated at each timestamp, $\mathcal{C}$ is composed of the timestamps where there is a change in the label from the previous one (e.g., stand $\to$ walk). A window $X_{t_m} = (\mathbf{x}_t)_{t=t_m-I}^{t_m+I-1}$ is a sequence of consecutive $2I$ points centered at timestamp $t_m$. A representation model $f_\theta$, which is a deep neural network parameterized by $\theta$, converts each window $X_{t_m}$ to its *representation* $\mathbf{z}_{t_m} \in \mathbb{R}^{d'}$, i.e., $\mathbf{z}_{t_m} = f_\theta(X_{t_m})$.

**Representation Learning:** RECURVE is not bound to a specific representation learning method, and we summarize the training process using one of the popular methods, the *temporal predictive coding (TPC)* proposed in TS-CP$^2$ (Deldari et al., 2021). Here, two non-overlapping consecutive windows are used as a positive pair, and two randomly-sampled windows are used as a negative pair. Thus, TS-CP$^2$ randomly samples $b$ windows as well as their succeeding windows and constructs a batch $B = \{X_{t_1}, X_{t_2}, \ldots, X_{t_b}, X_{t_1+2I}, X_{t_2+2I}, \ldots, X_{t_b+2I}\}$. Then, it minimizes the InfoNCE loss (Mnih & Kavukcuoglu, 2013),

$$\ell(B, \theta) = -\frac{1}{b} \sum_{j=1}^{b} \log \frac{\exp(\mathrm{sim}(\mathbf{z}_{t_j}, \mathbf{z}_{t_j+2I})/\tau)}{\sum_{k=1, k \neq j}^{b} \exp(\mathrm{sim}(\mathbf{z}_{t_j}, \mathbf{z}_{t_k})/\tau)}, \tag{1}$$

where $\text{sim}(\cdot, \cdot)$ is the cosine similarity function, $\exp(\cdot)$ is the exponential function, $\mathbf{z}_{t_j} = f_\theta(X_{t_j})$, and $\tau$ is a scaling parameter. The model parameter $\theta$ is updated iteratively by gradient descent, i.e., $\theta \leftarrow \theta - \eta \nabla_\theta \ell(B, \theta)$, where $\eta$ is a learning rate.

**Change Metric and Detection:** Using the representations of all windows centered at each point in $\mathcal{X}$, i.e., $\{\mathbf{z}_t \mid t \in [\![1, \ T]\!]\}^1$, a *change metric* $\hat{y}_t$ is derived for each point $\mathbf{x}_t \in \mathcal{X}$, which represents the probability that $\mathbf{x}_t$ is a change point. For example, the change metric in TS-CP$^2$ employs the *distance* (i.e., cosine similarity) between the embeddings of adjacent points,

$$\hat{y}_t^{\text{dist}} = \texttt{MinMaxNorm}(|\text{sim}(\mathbf{z}_t, \mathbf{z}_{t+1}) - \texttt{MovAvg}(\text{sim}(\mathbf{z}_t, \mathbf{z}_{t+1}))|), \qquad (2)$$

where $\texttt{MovAvg}$ calculates a simple central moving average and $\texttt{MinMaxNorm}$ is min-max normalization over all timestamps to rescale a value between 0 and 1. Then, similar to binary classification, the points whose change metric exceeds a certain threshold $\varphi$ are identified as change points,

$$\hat{\mathcal{C}} = \{t \mid \hat{y}_t \geq \varphi \text{ where } t \in [\![1, \ T]\!]\}. \qquad (3)$$

**Goal:** Obviously, an effective change metric is crucial to the success of change point detection. Therefore, we propose a novel change metric, $\hat{y}_t^{curv}$, using the *curvatures in the representation space* instead of the consecutive distances in the representation space.

## 3.2 Curvature-Based Change Metric

A trajectory usually refers to the path or track that an object (e.g., human and vehicle) in motion follows through space and time (Lee et al., 2007). Thus, we get to Definition 3.1 if we think of an object as a point floating in the representation space.

**Definition 3.1** (TRAJECTORY). *A **representation trajectory** (simply **trajectory**) $\mathcal{T}$ is a curve specified by a sequence of representations at consecutive timestamps and denoted as $\mathcal{T} = (\mathbf{z}_t)_{t=1}^{|\mathcal{T}|}$.*

The curvature at a specific point on a curve is the rate at which the direction of the curve changes instantaneously at the point (Lewiner et al., 2005). It is a well-defined concept in geometry and quantifies how sharply or gradually the curve bends or deviates from a straight line. We employ the definition designed for a trajectory (Buchin et al., 2011). For three timestamps in order, $t^-$, $t$, and $t^+$, where $t^- < t < t^+$, consider

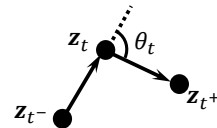

Figure 3: Turning angle.

their representations $\mathbf{z}_{t^-}$, $\mathbf{z}_t$, and $\mathbf{z}_{t^+}$. Two difference vectors, $\mathbf{z}_t - \mathbf{z}_{t^-}$ and $\mathbf{z}_{t^+} - \mathbf{z}_t$, are naturally derived, and the *turning angle* $\theta_t$ between them in Figure 3 is calculated by

$$\theta_t = \arccos \frac{(\mathbf{z}_t - \mathbf{z}_{t^-}) \cdot (\mathbf{z}_{t^+} - \mathbf{z}_t)}{||\mathbf{z}_t - \mathbf{z}_{t^-}|| ||\mathbf{z}_{t^+} - \mathbf{z}_t||}. \qquad (4)$$

Each value of $\theta_t$ ranges between 0 and $\pi$, where $t \in [\![2, \ T-1]\!]$. Then, the *curvature* is the rate of the direction changes between the two difference vectors, i.e. how much a difference vector rotates per unit length, as defined in Definition 3.2.

**Definition 3.2** (CURVATURE). *The **curvature** at timestamp $t$ in a representation trajectory $\mathcal{T}$ is the turning angle $\theta_t$ divided by the sum of the difference vector lengths,*

$$\kappa_t = \frac{\theta_t}{||\mathbf{z}_t - \mathbf{z}_{t^-}|| + ||\mathbf{z}_{t^+} - \mathbf{z}_t||}. \qquad (5)$$

According to our observations and intuitions described in Section 1, the curvature of an intra-segment point is higher than that of an inter-segment point. Thus, the curvature defined in Definition 3.2 can be used as a change metric. For stability, the timestamps $t^-$ and $t^+$ in Eq. (5) are determined to be $w > 1$ timestamps before and after timestamp $t$. We set $w$ to 5% of the mean segment length, which is observed to work well in most situations. Please refer to Section 4.4 about the sensitivity analysis on the value of $w$. Definition 3.3 concludes our novel curvature-based change metric.

**Definition 3.3** (CHANGE METRIC). *The **curvature-based change metric** at timestamp $t$ becomes*
$$\hat{y}_t^{curv} = \texttt{MovAvg}(1 - \texttt{MinMaxNorm}(\kappa_{t,w})), \qquad (6)$$
where $\kappa_{t,w}$ is obtained from Eq. (5) with $t^- = t - w$ and $t^+ = t + w$ ($w > 1$); and $\texttt{MovAvg}$ and $\texttt{MinMaxNorm}$ are the same as Eq. (2).

---

[1]Due to one-to-one correspondence between windows and points, each $\mathbf{z}_t$ can be regraded as the representation of each $\mathbf{x}_t$.

### 3.3 CHANGE METRIC THRESHOLDING

Once the change metric $\hat{y}_t^{curv}$ in Eq. (6) is prepared, it is possible to detect change points by finding the points where $\hat{y}_t^{curv} \geq \varphi$, as formulated by Eq. (3). Therefore, it is necessary to develop a heuristic for determining the threshold $\varphi$, and additional information can be utilized for this purpose. Such additional information includes the mean segment length and the validation dataset. If the mean segment length, i.e., the average of the lengths of segments distinguished by change points, is known, the estimated number of change points can be calculated by dividing the total number of timestamps by the mean segment length. The threshold $\varphi$ is then determined to obtain the estimated number of change points. Alternatively, if we have a validation dataset, we select the threshold $\varphi$ that yields the best performance based on an evaluation measure. Empirical evaluation in Section 4 employs the heuristic based on the mean segment length.

### 3.4 THEORETICAL ANALYSIS

Our theoretical analysis is conducted by showing the following properties: **(1)** the intra-segment points in the representation space are confined within a smaller hypersphere than the inter-segment points, as shown in Figure 4; **(2)** the mean total curvature of a representation trajectory increases as the radius of the confining hypersphere decreases, which leads to the rationale behind Definition 3.3.

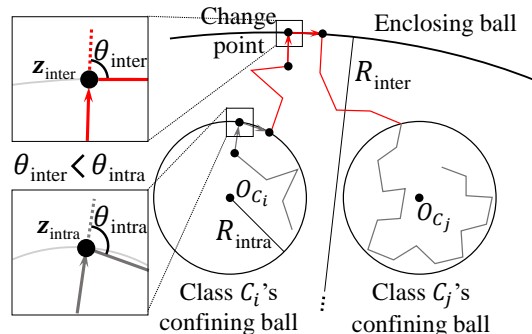

Figure 4: Comparison of the curvatures between intra- and inter-segment points.

**Proposition 3.4** (CONFINEMENT). *Consider a subsequence of a specific class $C_i$, $\mathcal{X}_{C_i} = (\mathbf{x}_t)_{t=t_{start}}^{t_{end}}$, as well as its representation trajectory, $\mathcal{T}_{C_i} = (\mathbf{z}_t)_{t=t_{start}}^{t_{end}}$, in Figure 4. Then, $\mathcal{T}_{C_i}$ is **confined** within a hypersphere $S_{C_i} \subset \mathbb{R}^{d'}$ centered at $O_{C_i} \in \mathbb{R}^{d'}$ of radius $R_{intra}$. That is, $\forall t \in [\![t_{start},\ t_{end}]\!]$, $\mathbf{z}_t \in S_{C_i}$ holds.*

Regarding the proof of Proposition 3.4, it is widely known that representation (contrastive) learning produces *class-separated* representations (Wang et al., 2022; Parulekar et al., 2023). According to Wang et al. (2022), the augmentations of positive examples overlap under some assumptions, and the positive examples form a connected graph based on augmentation overlap; thus, the alignment of positive examples by constrastive learning will cluster the examples of the same class together and lead to class-separated representations.

**Proposition 3.5** (CONFINEMENT RADIUS). *Consider a transition from a class $C_i$ to another class $C_j$ in Figure 4. Let $S_{C_i}, S_{C_j} \subset \mathbb{R}^{d'}$ be the confining hyperspheres for $C_i$ and $C_j$, respectively, of radius $R_{intra}$. Then, consider a larger hypersphere of radius $R_{inter}$ that encloses the inter-segment points (in red) as well as $S_{C_i}$ and $S_{C_j}$. Thus, $R_{intra} < R_{inter}$ holds by definition.*

Based on temporal coherence (Deldari et al., 2021; Shin et al., 2022; 2023) inherent in time series, we make an assumption on the representation trajectory before proceeding to the second step.

**Assumption 3.6** (EQUILATERAL RANDOM WALK). A representation trajectory $\mathcal{T} = (\mathbf{z}_t)_{t=1}^{|\mathcal{T}|}$ is a Markov chain, where $\mathbf{z}_t$ is sampled over the surface of the *unit* hypersphere centered at $\mathbf{z}_{t-1}$ and also contained in a confining hypersphere of radius $R$. That is, $||\mathbf{z}_t - \mathbf{z}_{t-1}|| = 1$ ($t \in [\![2, |\mathcal{T}|]\!]$) and $||\mathbf{z}_t|| < R$ ($t \in [\![1, |\mathcal{T}|]\!]$) such that $R > 1$.

Under Assumption 3.6, the curvature in Eq. (5) becomes the turning angle in Eq. (4) because the denominator is reduced to a constant. Then, when a given representation trajectory $\mathcal{T}$ is confined by a hypersphere of radius $R$, its *mean total curvature* is defined by

$$\mathcal{K}_\mathcal{T}(R) = \frac{1}{|\mathcal{T}|} \sum_{\mathbf{z}_t \in \mathcal{T}} \mathbb{E}_{\mathbf{z}_t|R}[\theta_t], \tag{7}$$

where $\mathbb{E}_{\mathbf{z}_t|R}[\theta_t]$ is the expectation of the curvature at timestamp $t$ with respect to the distribution of the representations in the confining hypersphere of radius $R$.

**Proposition 3.7** (MEAN TOTAL CURVATURE). *Consider a representation trajectory $\mathcal{T}$ confined in a hypersphere of radius $R$ under Assumption 3.6. Then, the mean total curvature $\mathcal{K}_\mathcal{T}(R)$ is a **decreasing** function of the radius $R$, i.e., $\frac{\mathrm{d}}{\mathrm{d}R}\mathcal{K}_\mathcal{T}(R) < 0$.*

The proof of Proposition 3.7 is provided by Diao et al. (2013). The mean total curvature is rigorously formulated as a complicated integral. By a simulation of random walk with one million steps, the decrease in the curvature is represented by the linear function $3.53 - 1.21R$ and the function $\pi/2 + 0.65/R^{1.5}$ for two different regimes of $R$.

**Notation.** The representation trajectories confined within the hyperspheres of radii $R_{intra}$ and $R_{inter}$ in Figure 4 are called *intra-segment* and *inter-segment* trajectories as well as denoted by $\mathcal{T}_{intra}$ and $\mathcal{T}_{inter}$, respectively.

Putting Propositions 3.5 and 3.7 together, the observation on the difference in the curvature is finally formalized by Theorem 3.8.

**Theorem 3.8** (CURVATURE DIFFERENCE). *The mean total curvature of an intra-segment trajectory is greater than that of an inter-segment trajectory, i.e., $\mathcal{K}_{\mathcal{T}_{intra}}(R_{intra}) > \mathcal{K}_{\mathcal{T}_{inter}}(R_{inter})$.*

*Proof.* Because $R_{intra} < R_{inter}$ by Proposition 3.5, $\mathcal{K}_{\mathcal{T}_{intra}}(R_{intra}) > \mathcal{K}_{\mathcal{T}_{inter}}(R_{inter})$ obviously holds by the decreasing nature of $\mathcal{K}_\mathcal{T}(R)$ of Proposition 3.7. $\square$

Theorem 3.8 can be intuitively explained if we consider the special case in which the next representation of $\mathbf{z}_{intra}$ or $\mathbf{z}_{inter}$ lies on the surface of a hypersphere, as visualized in Figure 4. Since a smaller radius necessitates a sharper turn, $\theta_{intra} > \theta_{inter}$ holds true. In this particular instance, where $\mathbf{z}_{intra}$ or $\mathbf{z}_{inter}$ is an orthogonal projection onto the surface, the turning angle can be expressed as $\pi - \arccos\frac{1}{2R}$, which is also a *decreasing* function of $R$.

### 3.5 EMPIRICAL PROOFS

The findings in the theoretical analysis also align well with the visualizations of the representations from a real dataset. Figure 5 displays three representation trajectories in the representation space of two principal components, which are obtained by the TPC method with $d' = 32$ for the mHealth dataset. Each representation trajectory includes 100 points centered at a change point. Inter-segment points within 5 timestamps from the change point are denoted by "×", while intra-segment points are denoted by "•". The color of each symbol indicates the value of our change metric—i.e., $1-$curvature. Obviously, inter-segment points have higher values of the change metric than intra-segment points. Interestingly, in Figure 5, the distance between two consecutive representations remains similar regardless of whether they are intra- or inter-segment points. This result reaffirms the existence of temporal coherence in the representation space, which could reduce the accuracy of *distance-based* change point detection methods. Moreover, it is evident that the representation trajectories of intra-segment points exhibit clearer confinement, resulting in more closed shapes and larger average turning angles. The representation trajectories of inter-segment points undergo fewer rotations and produce a relatively straighter shape.

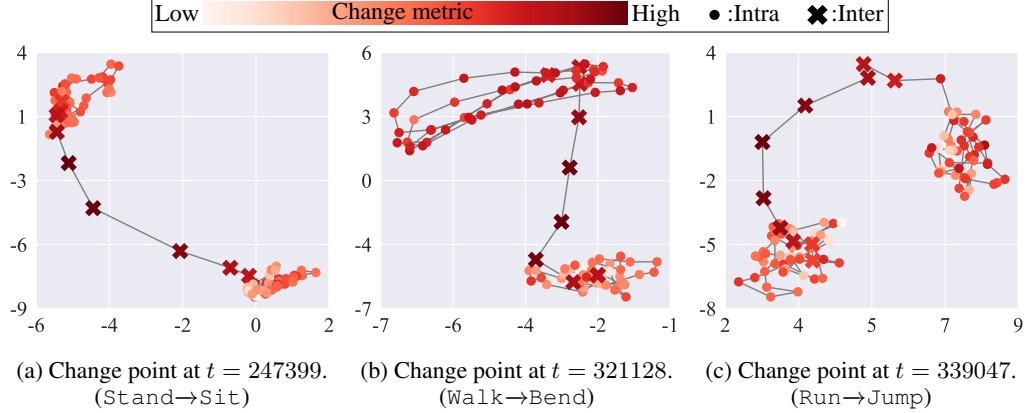

(a) Change point at $t = 247399$. (Stand→Sit)

(b) Change point at $t = 321128$. (Walk→Bend)

(c) Change point at $t = 339047$. (Run→Jump)

Figure 5: Three representation trajectories in the space of two principal components in mHealth.

## 4 EVALUATION

### 4.1 EXPERIMENT SETTING

**Datasets:** The profiles of the four datasets used in our experiments are summarized in Table 1, which lists the number of timestamps, mean segment length, number of classes, data dimensionality, sampling rate in Hz, and number of change points.

Table 1: Summary of datasets and hyper-parameters.

| Dataset | Timestamps | Length | #class | $d$ | Rate | #CP | Window | Epoch |
|---------|-----------|--------|--------|-----|------|-----|--------|-------|
| WISDM | 343092 | 697 | 6 | 3 | 20 | 491 | 50 | 10 |
| HAPT | 407807 | 903 | 6 | 6 | 50 | 450 | 100 | 50 |
| mHealth | 343195 | 2932 | 12 | 23 | 50 | 119 | 100 | 50 |
| 50salads | 496250 | 551 | 19 | 2048 | 30 | 898 | 50 | 100 |

WISDM (Kwapisz et al., 2011), HAPT (Anguita et al., 2013), and mHealth (Anguita et al., 2013) are human action recognition datasets, which are measured by single or multiple accelerometers and/or gyroscopes. 50salads (Stein & McKenna, 2013) is a video dataset that captures 25 people preparing salads; the I3D features of 2048 dimensions are extracted, following Farha & Gall (2019). The set of ground-truth change points, $\mathcal{C}$, is defined as the set of the timestamps where the class changes. The dimensionality of the representation space is set to $d' = 8$ for WISDM and HAPT and $d' = 32$ for mHealth and 50salads, considering their data dimensionality.

**RECURVE Details:** To obtain the point representations, we employ two time-series representation learning methods, TPC proposed in TS-CP$^2$ (Deldari et al., 2021) and TNC (Tonekaboni et al., 2021). RECURVE+TPC and RECURVE+TNC indicate the two implementations depending on the representation learning method. A temporal convolutional network (TCN) is trained in both methods. Note that any representation learning method can be combined with RECURVE. The window size, $2I$, and the number of training epochs for each dataset are shown in Table 1, where the window size is approximately twice the sampling rate. The learning rate is set to $0.005$ for all datasets. The hyperparameter $w$, indicating the length of a representation vector, is set to $5\%$ of the mean segment length. A moving average in Eq. (6) is computed using the ten timestamps preceding and following each timestamp. RECURVE is implemented using PyTorch 1.13.0, and its source code is available at `https://bit.ly/3ET7vmg`.

**Compared Methods:** RuLSIF (Hushchyn & Ustyuzhanin, 2021), KL-CPD (Chang et al., 2019), and TS-CP$^2$ (Deldari et al., 2021) are chosen as the representative method from each of the three categories in Section 2.1. The window size in Table 1 is applied to all compared methods for fair comparison. A multilayer perceptron is used for the regressor of RulSIF. The hyperparameters of RuLSIF and KL-CPD are favorably determined by a grid search, as detailed in Appendix A. The public implementations of RuLSIF$^2$ and KL-CPD$^3$ are used for our experiments. TS-CP$^2$ is the closest to our work, and its main mechanism is briefly described in Section 3.1. Because representation learning itself is shared between TS-CP$^2$ and RECURVE when TPC is used, the same hyperparameter setting is applied to both methods whenever possible. TS-CP$^2$ is re-implemented using PyTorch 1.13.0 for direct comparison with RECURVE.

**Evaluation Measures:** First, the Area Under the ROC Curve (AUC) is measured by considering change point detection as binary classification with a binary label vector $\mathbf{y} \in \{0, 1\}^T$ converted from $\mathcal{C}$. Following Deldari et al. (2021), an error margin is introduced to accommodate some noise from annotation and detection. A detected change point is considered to be correct if it lies within $p$ timestamps from one of the ground-truth change points. For this purpose, $\mathbf{y}$ is relaxed to

$$y_t = \begin{cases} 1 & \text{if } t_k - p \leq t < t_k + p \text{ where } t_k \in \mathcal{C} \\ 0 & \text{otherwise.} \end{cases} \tag{8}$$

Then, for $t \in [\![1, T]\!]$, whether ($\hat{y}_t$ in Eqs. (2) or (6) $\geq \varphi$) is compared against $y_t$ in Eq. (8). We use multiple error margins, $p \in \{5, 10, 20\}$, since a margin could be different for diverse applications (Aminikhanghahi & Cook, 2017). Second, the mean LOCation distance (LOC) is measured, which is the average distance from a detected change point to its closest ground-truth change point (Gharghabi et al., 2019; Schäfer et al., 2021). The LOC measure is useful for checking the preciseness of the change points detected.

Regarding the threshold $\varphi$, the AUC measure does not require a specific value because it evaluates the true positive and false positive rates over a given range. For the LOC measure, two values are

---

$^2$`https://github.com/HSE-LAMBDA/roerich/tree/main`
$^3$`https://github.com/HolyBayes/klcpd`

used for each experiment: one is determined to achieve the best F1 score, and the other is determined by the heuristic based on the mean segment length in Section 3.3, where the estimated number of change points is multiplied by $p = 10$, taking the error margin into account.

For each evaluation measure, we conduct every experiment *five* times with different seeds and report the average as well as the standard deviation.

## 4.2 COMPARISON WITH STATE-OF-THE-ART METHODS

Tables 2 and 3 display the AUC and LOC measures for the five methods across the four datasets. The AUC measure is presented in Table 2 with varying the error margin $p$. RECURVE outperforms the other change point detection methods, where the optimal representation approach varies for each dataset. RECURVE wins against TS-CP$^2$ in *all* datasets, irrespective of the evaluation measure. This finding demonstrates that the curvature is more effective for change point detection in temporally coherent time series where the class changes gradually. WISDM, HAPT, and mHealth exhibit

Table 2: Overall change point detection accuracy in the AUC measure (the best results **in bold**).

| Methods | $p$ | AUC ↑ | | | |
|---|---|---|---|---|---|
| | | WISDM | HAPT | mHealth | 50salads |
| RuLSIF | 5 | 0.559±0.005 | 0.797±0.001 | 0.598±0.002 | 0.606±0.006 |
| | 10 | 0.560±0.005 | 0.797±0.001 | 0.599±0.002 | 0.608±0.003 |
| | 20 | 0.563±0.005 | 0.797±0.001 | 0.600±0.002 | 0.611±0.004 |
| KL-CPD | 5 | 0.697±0.000 | 0.868±0.003 | 0.842±0.117 | 0.682±0.003 |
| | 10 | 0.702±0.000 | 0.873±0.004 | 0.849±0.113 | 0.684±0.003 |
| | 20 | 0.710±0.000 | 0.875±0.005 | 0.856±0.105 | 0.689±0.003 |
| TS-CP$^2$ | 5 | 0.815±0.012 | 0.692±0.007 | 0.560±0.014 | 0.680±0.010 |
| | 10 | 0.820±0.012 | 0.695±0.006 | 0.561±0.013 | 0.682±0.009 |
| | 20 | 0.823±0.013 | 0.697±0.006 | 0.561±0.010 | 0.685±0.008 |
| **RECURVE** **+TPC** | 5 | **0.897±0.003** | **0.909±0.001** | 0.954±0.003 | **0.719±0.005** |
| | 10 | **0.901±0.004** | **0.913±0.001** | 0.954±0.003 | **0.723±0.006** |
| | 20 | 0.902±0.003 | **0.919±0.001** | 0.955±0.005 | **0.729±0.006** |
| **RECURVE** **+TNC** | 5 | 0.880±0.004 | 0.863±0.017 | **0.979±0.004** | 0.594±0.016 |
| | 10 | 0.889±0.004 | 0.867±0.017 | **0.980±0.004** | 0.595±0.016 |
| | 20 | **0.905±0.004** | 0.876±0.017 | **0.980±0.005** | 0.600±0.015 |

periodicity in certain classes, including walking and running. This periodicity would produce a closed shape for intra-segment trajectories and increase their curvatures, enhancing the performance of RECURVE. In particular, when $p = 20$, RECURVE outperforms the second-best method by up to 12.7% in terms of the AUC measure for the mHealth dataset.

Table 3: Overall change point detection accuracy in the LOC measure (the best results **in bold**).

| Methods | LOC ↓ (thresholding by best F1) | | | | LOC ↓ (thresholding by mean segment length) | | | |
|---|---|---|---|---|---|---|---|---|
| | WISDM | HAPT | mHealth | 50salads | WISDM | HAPT | mHealth | 50salads |
| RuLSIF | 420.9±18.54 | 108.2±0.188 | 780.0±8.580 | 184.4±1.463 | 429.5±9.968 | 156.0±0.092 | 802.6±30.18 | 189.2±1.120 |
| KL-CPD | 189.0±12.20 | 121.5±4.540 | 306.4±126.5 | 179.5±3.853 | 198.3±2.329 | 113.0±2.545 | 352.6±119.7 | 176.6±1.017 |
| TS-CP$^2$ | 166.6±7.840 | 386.6±31.04 | 879.4±62.57 | 119.0±6.712 | 183.1±15.13 | 404.2±32.60 | 923.8±44.39 | 129.4±5.091 |
| **RECURVE+TPC** | **114.7±56.07** | **33.25±1.290** | 483.6±64.24 | **79.29±10.52** | **178.4±36.05** | **34.28±0.727** | 341.0±47.93 | **93.76±7.475** |
| **RECURVE+TNC** | 210.0±112.3 | 47.92±2.884 | **224.0±211.2** | 175.0±26.38 | 219.8±102.2 | 50.71±1.589 | **239.6±212.4** | 178.8±20.87 |

## 4.3 DETAILED INVESTIGATION ON CHANGE METRIC QUALITY

We display the average values of the change metrics separately for each pair of classes using the HAPT dataset, which was chosen for ease of visualization due to its small number of classes. Figure 6a depicts the *inter-class embedding distance*, which is determined by the Euclidean distance between the centroids of point representations of given classes. The values of the change metrics are averaged across the *inter-segment* points for each distinct class transition. Figures 6b and 6c

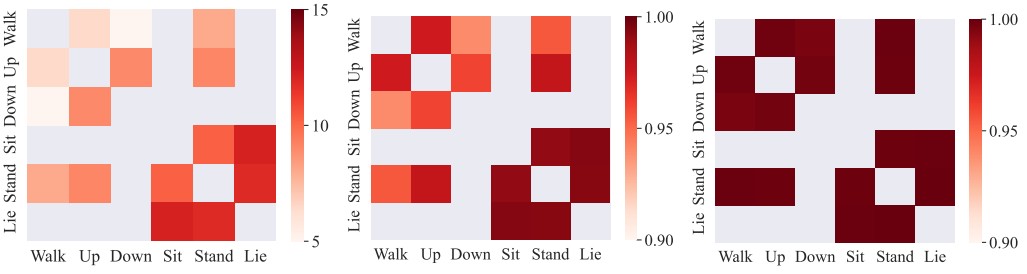

(a) Inter-class embedding distance.  (b) Averaged $\hat{y}_t^{\text{dist}}$ (TS-CP$^2$).  (c) Averaged $\hat{y}_t^{\text{curv}}$ (**RECURVE**).

Figure 6: Heatmaps of the inter-class distances and values of the change metrics between the classes in the HAPT dataset. A grey box indicates no transition between two classes.

are obtained by the *distance-based* change metric $\hat{y}_t^{\text{dist}}$ of TS-CP² and the *curvature-based* change metric $\hat{y}_t^{\text{curv}}$ of RECURVE, respectively. Intriguingly, $\hat{y}_t^{\text{curv}}$ generates high values for *all* class pairs in Figure 6c, which indeed explains the overall high accuracy in Tables 2 and 3. In contrast, in Figure 6b, $\hat{y}_t^{\text{dist}}$ only generates high values when the inter-class embedding distance is sufficiently large (i.e., abrupt change), whereas it generates moderate values when the inter-class embedding distance is small (i.e., gradual change). That is, Figures 6a and 6b show a very high correlation. In summary, $\hat{y}_t^{\text{curv}}$ is insensitive to the degree of changes whereas $\hat{y}_t^{\text{dist}}$ is not. Therefore, this result demonstrates the superiority of the curvature-based change metric over the distance-based change metric.

Figure 7 magnifies six class pairs selected from all class pairs depicted in Figures 6b and 6c. For example, Stand→Sit and Lie→Stand are accompanied by rapid body movement, and both TS-CP² and RECURVE capture the change points well, as evidenced by the high density in the interval close to 1. In contrast, when two action classes are comparable, as in Stand→Walk, Down→Up, and Walk→Down, the values of the change metric of TS-CP² disperse to other intervals, resulting in a decrease in detection performance. RECURVE maintains the same shape in all density plots due to the remarkable effectiveness of our curvature-based change metric.

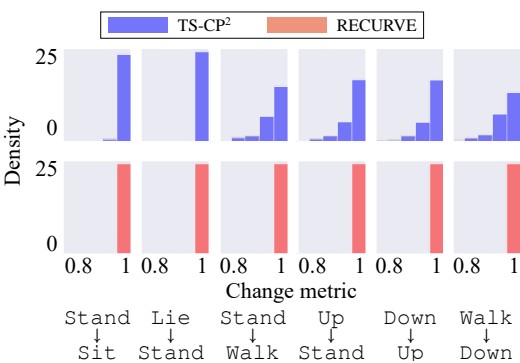

Figure 7: Distribution of the change metrics for each class transition in the HAPT dataset.

### 4.4 SENSITIVITY ANALYSIS ON REPRESENTATION VECTOR LENGTH $w$

Table 4 shows the performance of RECURVE while varying the representation vector length $w$ (see Definition 3.3) when the error margin $p$ for the AUC measure is fixed at 10. The value of $w$ ranges from $0.25\times$ to $4.00\times$ of the default value, which is set to $5\%$ of the mean segment length (indicated by $1.00\times$). If the value of $w$ were too large, the denominator of Eq. (5) would be too large for any point in a time series, and the curvature would be unable to distinguish between intra- and inter-segment points. If the value of $w$ were too small, some noise in point representations would distort the curvature. Under this trade-off, the default value performs the best in terms of both evaluation measures when it is averaged over the four datasets and the two representation learning methods. On a dataset with lengthy segments, such as mHealth, the sensitivity tends to decrease, and there is small variation when varying the value of $w$.

Table 4: Performance of RECURVE with varying the hyperparameter $w$ (the best results **in bold**).

| Repr. | $w$ | AUC ↑ | | | | LOC ↓ (thresholding by mean segment length) | | | |
|---|---|---|---|---|---|---|---|---|---|
| | | WISDM | HAPT | mHealth | 50salads | WISDM | HAPT | mHealth | 50salads |
| TPC | 0.25× | 0.832±0.015 | 0.901±0.004 | 0.911±0.008 | 0.685±0.007 | 358.7±89.01 | 40.41±2.233 | 654.7±36.04 | 136.9±2.064 |
| | 0.50× | 0.891±0.006 | **0.914±0.002** | 0.953±0.004 | 0.703±0.005 | 246.3±139.6 | 37.42±1.309 | 538.9±49.69 | 120.0±2.127 |
| | 1.00× | **0.901±0.004** | 0.913±0.001 | **0.954±0.003** | **0.723±0.006** | **178.4±36.05** | **34.28±0.727** | 341.0±47.93 | **93.76±7.475** |
| | 2.00× | 0.892±0.003 | 0.887±0.001 | 0.927±0.004 | 0.692±0.004 | 252.9±97.20 | 42.74±5.383 | 821.2±53.60 | 94.21±6.153 |
| | 4.00× | 0.861±0.002 | 0.847±0.002 | 0.893±0.004 | 0.604±0.004 | 273.8±119.0 | 53.00±10.77 | 628.2±39.36 | 104.0±3.502 |
| TNC | 0.25× | 0.824±0.016 | 0.842±0.011 | 0.956±0.009 | 0.580±0.015 | 249.7±37.59 | 52.95±3.698 | 213.7±110.2 | 222.0±7.443 |
| | 0.50× | 0.869±0.009 | 0.850±0.012 | 0.978±0.005 | 0.587±0.014 | 231.3±79.25 | 51.15±2.685 | **236.4±136.5** | 218.9±16.16 |
| | 1.00× | 0.889±0.004 | **0.867±0.017** | **0.980±0.004** | **0.595±0.016** | 219.8±102.2 | **50.71±1.589** | 239.6±212.4 | **178.8±20.87** |
| | 2.00× | **0.897±0.002** | 0.827±0.019 | 0.962±0.007 | 0.583±0.008 | **196.0±79.98** | 58.19±1.588 | 346.6±305.9 | 179.4±16.22 |
| | 4.00× | 0.871±0.002 | 0.773±0.016 | 0.937±0.007 | 0.568±0.009 | 265.5±58.16 | 97.38±6.328 | 265.6±92.20 | 183.2±17.64 |

Additional sensitivity analysis on the representation dimensionality $d'$ is available in Appendix B.

## 5 CONCLUSION

In this paper, we present RECURVE, a novel change-point detection method that uses the curvature of a representation trajectory to replace the consecutive distance for a change metric. Theoretically, the mean total curvature of an intra-segment trajectory is greater than that of an inter-segment trajectory due to the confining nature of the representations of the points within a single class. Unlike the consecutive distance, this property of the curvature is insensitive to the degree of the changes between two classes (segments). Our comprehensive experiments confirm that RECURVE achieves up to 12.7% higher detection accuracy than state-of-the-art methods. Overall, we believe that our work opens the door to a new direction for change point detection in time series.

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

## A    HYPERPARAMETERS FOR COMPARED METHODS

For RuLSIF, we conduct a grid search for the learning rate (LR) $= \{0.05, 0.1, 0.2\}$, the weight of L2 normalization $\lambda_{\text{L2}} = \{0.01, 0.05, 0.1\}$, and the parameter of the RuLSIF loss $\alpha = \{0.01, 0.05, 0.1\}$. When applying RuLSIF to the four datasets, we use a multilayer perceptron with a single hidden layer with 100 units and train it with a batch size of 32 for 50 epochs. For KL-CPD, we conduct a grid search to determine the optimal hidden dimensionality $h = \{10, 50, 100\}$ of the RNN encoder/decoder, as well as the values for the hyperparameters $\lambda_{\text{AE}} = \{0.1, 0.01, 0.001\}$ and $\lambda_{\text{Real}} = \{0.1, 0.01, 0.001\}$ which govern the influence of the reconstruction loss and the MMD2 loss on real datasets. For training the generator of KL-CPD, the batch size is set to 64, the number of epochs is set to 3, and the learning rate is set to 0.001. Table 5 provides a summary of the determined hyperparameter values.

Table 5: Hyperparameter values of RuLSIF (left half) and KL-CPD (right half) after a grid search.

| Dataset | LR | $\lambda_{\text{L2}}$ | $\alpha$ | $\lambda_{\text{AE}}$ | $\lambda_{\text{Real}}$ | $\#hidden$ |
|---|---|---|---|---|---|---|
| WISDM | 0.05 | 0.1 | 0.01 | 0.01 | 0.001 | 10 |
| HAPT | 0.2 | 0.01 | 0.01 | 0.01 | 0.1 | 10 |
| mHealth | 0.2 | 0.1 | 0.01 | 0.01 | 0.01 | 10 |
| 50salads | 0.05 | 0.01 | 0.05 | 0.1 | 0.01 | 50 |

## B    SENSITIVITY ANALYSIS ON REPRESENTATION DIMENSIONALITY

Table 6 shows the performance of RECURVE while varying the representation dimensionality $d'$ when the error margin $p$ for the AUC measure is fixed at 10. The value of $d'$ ranges from $0.25\times$ to $4.00\times$ of the default value, which is 8 for WISDM and HAPT or 32 for mHealth and 50salads (indicated by $1.00\times$). A trade-off point in the representation dimensionality exists for nearly all datasets. A representation space with an excessively high dimensionality is susceptible to the curse of dimensionality. If the value of $d'$ were too large, the turning angle and distance in Eq. (5) would be indistinguishable across all timestamps in a time series, as any two points in a high-dimensional space would become nearly orthogonal and their distance would always be similar. If the value of $w$ were too small, low-quality features would be extracted from the original time series by representation learning; thus, the performance degrades with an insufficient dimensionality as shown in the result of 50salads whose data dimensionality is 2048. Overall, the default setting is suitable for achieving competitive performance for all datasets.

Table 6: Performance of RECURVE with varying the hyperparameter $d'$ (the best results **in bold**).

| Repr. | $d'$ | AUC ↑ | | | | LOC ↓ (thresholding by mean segment length) | | | |
|---|---|---|---|---|---|---|---|---|---|
| | | WISDM | HAPT | mHealth | 50salads | WISDM | HAPT | mHealth | 50salads |
| TPC | 0.25× | 0.870±0.007 | 0.844±0.011 | 0.942±0.007 | 0.719±0.006 | 349.9±32.99 | 307.0±30.03 | 553.4±69.70 | 97.64±6.153 |
| | 0.50× | **0.906±0.003** | 0.836±0.168 | 0.949±0.006 | 0.719±0.007 | 377.5±443.7 | 104.4±130.3 | 586.1±39.86 | 97.33±5.553 |
| | 1.00× | 0.901±0.004 | **0.913±0.001** | **0.954±0.003** | **0.723±0.006** | **178.4±36.05** | **34.28±0.727** | **341.0±47.93** | **93.76±7.475** |
| | 2.00× | 0.882±0.017 | 0.905±0.003 | 0.942±0.005 | 0.718±0.005 | 180.6±59.32 | 35.62±2.669 | 600.9±36.82 | 100.9±3.814 |
| | 4.00× | 0.857±0.015 | 0.900±0.004 | 0.937±0.006 | 0.719±0.007 | 200.4±55.87 | 38.27±1.191 | 592.3±64.23 | 100.0±6.432 |
| TNC | 0.25× | 0.838±0.046 | 0.862±0.006 | 0.963±0.013 | 0.561±0.018 | 168.1±79.54 | 50.37±4.293 | 241.0±30.85 | 198.7±20.14 |
| | 0.50× | 0.882±0.008 | 0.859±0.011 | 0.971±0.007 | 0.570±0.013 | **149.5±35.37** | **49.17±1.943** | 245.0±125.0 | 198.7±37.32 |
| | 1.00× | **0.889±0.004** | 0.867±0.017 | **0.980±0.004** | 0.595±0.016 | 219.8±102.2 | 50.71±1.589 | **239.6±212.4** | **178.8±20.87** |
| | 2.00× | 0.877±0.006 | 0.875±0.009 | 0.972±0.003 | 0.581±0.011 | 290.4±148.8 | 55.35±1.800 | 260.7±51.26 | 215.6±15.55 |
| | 4.00× | 0.880±0.003 | **0.887±0.004** | 0.973±0.003 | **0.607±0.012** | 257.9±81.69 | 57.72±1.010 | 280.5±94.19 | 179.3±14.75 |

