# OpenReview forum: "Detecting Change Points in Time Series via Curvatures of Representation Trajectories"
_ICLR.cc/2024/Conference — ICLR 2024 Conference Withdrawn Submission_

### Official Review · Reviewer_aWE7 · 2023-10-13

**Soundness:** 2 fair
**Presentation:** 2 fair
**Contribution:** 2 fair
**Rating:** 3
**Confidence:** 4

**Summary:**

The paper is concerned with detection of changes, both gradual and abrupt, in high-dimensional time-series. It proposes to use the notion of curvature (second derivative/difference) of the representations (generated by a trained neural network) rather than distances between representations. Performance results and comparisons on 4 datasets are presented. Some theoretical results are stated.

**Strengths:**

The paper addresses an important problem: detection of changes which may be abrupt or gradual and for cases in which pre-change or post-change pdf/pmf may not be available (or tractable). The paper is reasonably well-organized. Proposed algorithm is compared with a few existing approaches on 4 ‘real’ datasets. Code was provided.

**Weaknesses:**

The presented theoretical results are weak, underlying assumptions are not stated, and they do not address the detectability of a change. The lack of experiments on synthetic data is a major limitation. It is not clear how one would use the proposed approach in an unsupervised setting, as would be the case with gradual change. The paper provides no validation of the claim that the proposed approach can detect gradual changes as well as abrupt changes. The experimental section lacks details.

**Questions:**

1) State-of-the-art discussion:
The discussion of the statistical approaches to change point detection is superficial. There is a very large and significant body of work that goes beyond simple CUSUM, and beyond assuming that pre- and post-change distributions are known.

2) Weak theoretical results:
- Multiple propositions are given, but the proof outlines are not convincing. Full proofs were not provided. What assumptions were made for Prop 3.4? Are they applicable to the problem at hand?

- The assumption that the representation is a Markov chain and resides on the surface of a hypersphere is a strong assumption and needs justification.

- Is Proposition 3.7 in Dialo et al.? If so, credit it as part of the proposition, not in a succeeding paragraph. The statement ‘By a simulation of a random walk …’ is not clear; is that in Dialo et al.? Or did the authors conduct the simulation? If the latter, provide details of what was simulated; what do the two different regions correspond to, and why.

- Useful theoretical results for change detection would be of the form of an expression that shows that the PFA (or PMD) is monotonically decreasing with some parameter (presumably curvature here) keeping other things fixed. The theoretical results in the paper do not offer any such guarantees. The paper states a proposition, that under unstated assumptions, indicates that the mean curvature is lower near change points – this is insufficient to establish that the change point can be detected with a given measure of confidence.

- How does curvature depend upon the choice of representation? This is a key question that has not been addressed. In other words, is curvature an intrinsic feature of the data? Are representations resulting in large curvature more / less susceptible to adversarial attacks?

- Minor but important fix: ‘Empirical proof’ is wrong; it could be ‘Empirical validation’. You cannot ‘prove’ things empirically.

3) The experimental setting is poorly explained:

-  It is nice to see results on ‘real’ datasets, but given the lack of control over the data, it is hard to discern what the proposed algorithm has ‘learned’, and where it might have difficulties. This is a significant shortcoming. In the classical approach to change detection, performance results are always presented on synthetic datasets so as to obtain insights - What percentage of the datasets were used for training, validation, and testing. Are the presented results on training data or on test data?

- It is good that the authors used multiple runs and provide both means and standard deviations. But computing a standard deviation on a sample size of 5 is not meaningful. All of the standard deviations reported in Table 2 are very small: the mean to standard-deviation ratio is about 100:1. This raises concerns about the randomness across the experiments. It is unusual to see such large mean-to-std ratios.

- AUC metrics are gross measures. Typically, one fixes the false alarm rate (at say 0.01 or 0.005) and then reports the probability of detection. This needs to be done in addition to the AUC.  How many changepoints were correctly identified within the prescribed p window value?

- The discussion of Figure 6 seems to indicate that the same results would be obtained from TS-CP^2 by replacing all its non-zero metrics by unity (or choose a threshold to fix it to 0/1). Why do we need curvature?

- The smallness of the intra-class distances in Fig 6.a does not indicate that the change was gradual.  The paper makes claims about detecting ‘gradual’ changes, but nothing in the numerical section points to a validation of this claim. Again, some experiments on synthetic data would have been insightful.

- In Sec 4.1: “The window size is approximately twice the sampling rate”: what sampling rate? Why is it “fair” to use the same window size for all the methods? Table 4 indicates that the choice of window size has a significant effect on the performance of Recurve – presumably the same holds for the other methods; so, the ‘fair’ claim is not apparent.

- What do ‘Down  Up’ and ‘Walk  Down’ transitions represent?

- (Minor) Table 2 – Recurve + TPC. The p=20 results are better than the p=10 which is shown in bold

4) Algorithm Details:

- The paper does not provide a specific sequence of steps that one should take to implement the approach on some other data set. What are the guidelines for choosing thresholds when ground truth is not known? How would the approach work in an unsupervised setting?

- The choice of positive and negative training pairs needs justification. Two consecutive segments need not belong to the same class, and two randomly chosen segments need not belong to different classes. Given that the proposed method is supervised, surely better performance would be obtained if the positive pair is from the same class, and the negative pair from different classes.

- Before equation (2): Please explain why the change metric is a ‘probability’. How was that derived? What assumptions were mad? The mere fact that it is non-negative and sums to 1 does not make it a probability.

- The normalization in equation (5) needs justification. If the points t-, t, and t+ are in the same "segment", the denominator will be small, and the curvature large. Is that desired? In any case, the paper later removes the normalization. Why introduce it? The angle by itself is intuitive.

- The notion of ‘w’ timestamps in the para after (5) is confusing (although this becomes clearer in the experiments section). The \kappa_{t,w} could have been simply defined in (5).

-  Minor: Why is a window of 2I points chosen and claimed to be centered; wouldn’t a window of 2I-1 or 2I+1 be better?

---

### Official Review · Reviewer_a2DN · 2023-10-25

**Soundness:** 3 good
**Presentation:** 3 good
**Contribution:** 3 good
**Rating:** 6
**Confidence:** 3

**Summary:**

In this paper the authors study a new metric for detecting change points in a time series which can be a preprocessing step or a task of its own. They propose the use of curvature of representation trajectories. First, the representation of each point in time (using sliding windows) of the time-series is computed. The second step, where this method differs from previous methods, is to see this series of time point representations as a trajectory in representation space and to compute the curvature of this trajectory. Then, unlike existing methods, the curvature of the representation is used as a change point metric to threshold over and classify time-series points into two classes, viz., change point and not a change point. Further, the properties of this metric are analysed theoretically to support the usage of the metric for detecting change points. Experiments establish superiority against chosen baselines.

**Strengths:**

1.  The paper introduces a novel metric for detecting change points in time series data, which distinguishes it from existing methods. The use of curvature in representation trajectories as a key component of the proposed method is an innovative approach.

2.  The paper offers an in-depth theoretical analysis of the proposed metric, providing a solid foundation for its application in change point detection. This theoretically supports the credibility and reliability of the metric.

3.  While there is room for improvement, the paper maintains a good level of clarity in its presentation. The paper defines the problem of change point detection in a concise and well-structured manner, and conveys complex concepts and techniques in a comprehensible manner making it easy for readers to understand.

4.  The paper outlines the contributions and relevance of the work in the field of time series analysis supported by experiments against strong existing methods such as TS-CP2, RuLSIF, and KL-CPD along with detailed investigation on change metric quality averaged across inter-segment points for distinct class transitions.

5.  The paper provides sufficient details and information on the method, making it possible to reproduce the results.

**Weaknesses:**

1. **Lack of Clarity in Figure Usage:** The paper’s use of Figure 2 to explain the idea behind curvature-based change point detection might be challenging. As curvature is an abstract concept, it is difficult to comprehend the figure’s relevance without prior theoretical context. Suggestion: postpone the figure and its explanation until after the theoretical section to enhance clarity.

2. **Incomplete Proofs:** The paper references proofs from other works and, in some instances, lacks complete proofs (e.g., Proposition 3.5). This can be confusing and frustrating for readers, as it necessitates navigating multiple papers non-linearly to understand the method fully. A suggestion is to place complete proofs in an appendix with consistent notation to provide a comprehensive and self-contained resource.

3. **Notation Ambiguity:** The use of double brackets to represent ranges in mathematical notation, without prior explanation, deviates from the conventional mathematical convention of single brackets. This deviation can create confusion and uncertainty regarding the intended meaning of the notation. Providing a clear explanation or adhering to standard notation would improve clarity.

4. **Complex Definition without Adequate Explanation:** Definition3.3introducesthecurvature-basedchange metric, which comprises several sub-parts, including a moving average function, a min-max norm, and a subtraction (the “1−” in $MovAvg(1 − MinMaxNorm(\kappa_{t,w}))$. However, the paper lacks a detailed breakdown and explanation of these components. The origin of the "one minus" and the "moving average" is unclear, and the rationale behind this definition is not readily apparent. Enhancing the presentation by providing a step-by-step breakdown and connecting it to the theoretical foundation would improve understanding.

**Questions:**

1. I understand what $|sim(\mathbf{z}_t,\mathbf{z}_t+1) − MovAvg(sim(\mathbf{z}_t,\mathbf{z}_t+1))|$ is representing in eqn. (2). However, I do not have an intuitive or theoretical understanding of MovAvg(1 − MinMaxNorm(κt,w). Since moving average is the average summary of a moving window, is the change point metric a summary of the curvatures seen in the current window? Why is that a good change point metric? Please clarify this.

2. Why has the method not been compared against Espresso [1], aHSIC [2], and FLOSS [3]?

3. Although this has already been mentioned, I’ll use this section to highlight this again. Some propositions have not been proved, and the proof is outsourced to other papers (Diao et al. (2013)) for example. I believe these should be included in the appendix for completeness, because it’s hard to identify which propositions have been proved, which have been left for other works, and which have not been proved at all.

---

### Official Review · Reviewer_M4yt · 2023-11-01

**Soundness:** 3 good
**Presentation:** 3 good
**Contribution:** 2 fair
**Rating:** 5
**Confidence:** 4

**Summary:**

The paper presents RECURVE, a change-point detection solution that relies on the curvature changes of the representation trajectory. The idea is to eliminate problems when considering consecutive distances for a change metric. Experimental results across different datasets demonstrate the potential of the proposed solution.

**Strengths:**

- Change point detection is a well-studied and challenging problem
- The idea seems technically sound, supported by theoretical and practical proofs.
- Experimental results support the core claims of this work

**Weaknesses:**

- Novelty is somewhat low
- Missing baselines
- Lacking strong motivation for the proposed solution

**Questions:**

- Novelty is somewhat low

The approach heavily relies on earlier solutions. All "sub-components" of the work more or less exist. They are nicely integrated, so the solution appears technically sound, but the novelty is low (other than a combination of earlier results). (e.g., the technique relies on TS-CP2, then adopts a new loss function that already exists, the notion of curvature, and several of these definitions already exist and they are adopted for this application.)

- Missing baselines

The paper covers several relevant baselines but focus on a small DNN subset of the literature. A convincing work needs comparisons with several of the methods mentioned in the related work (i.e., non-DNN methods).

- Lacking strong motivation for the proposed solution

The examples used for motivation are somewhat "toy" cases. Real-world scenarios would strengthen the motivation for this work

---

### Official Review · Reviewer_MRxh · 2023-11-02

**Soundness:** 1 poor
**Presentation:** 3 good
**Contribution:** 2 fair
**Rating:** 3
**Confidence:** 3

**Summary:**

This article describes an approach to detect change-points with a novel measure called the trajectory curvature. The proposed metric seems natural and is well described. The theoretical analysis is less convincing. Experiments are thorough and compelling.

**Strengths:**

The experiments are thorough and convincing. The proposed method is applied to several data sets and compared with state-of-the-art approaches.

**Weaknesses:**

The theoretical analysis is unclear and should be rewritten to highlight the assumptions better and make the claims more rigorous.

**Questions:**

Most of my questions are about Section 3.4: Theoretical analysis.

- Proposition 3.4 is given without any hypothesis on the data or how the trajectories are estimated. Certainly, the trajectories are learned by minimizing the infoNCE, but we can only guess. Also, it is not clear how the class membership is useful. The authors provide references, but it is not clear how to derive or understand the proposition using those references.

- Does Proposition 3.5 state that there always is a ball that can contain two hyperspheres?

- Is there any reason to believe that Assumption 3.6 holds? Is the constraint on the increments ($||z_t - z_{t-1}||=1$) enforced in practice?

- The authors write, "[...] the denominator is reduced to a constant." below Assumption 3.6. Unless I misunderstand, this is only true if w=1. In addition, in Definition 3.3, the authors assume w>1.

- For Proposition 3.7, again, the authors only provide a reference to an article, which makes it difficult to verify the claim. Besides, the referenced article only deals with 2D and 3D random walks (I skimmed through it).

Minor comments:
- Inconsistent use of "change-point" or "change-point."
- In the experiments, what are inter-segment points? Is there a margin parameter?